# Stress Adaptation Responses of a *Listeria monocytogenes* 1/2a Strain via Proteome Profiling

**DOI:** 10.3390/foods12112166

**Published:** 2023-05-27

**Authors:** Federica D’Onofrio, Maria Schirone, Antonello Paparella, Ivanka Krasteva, Manuela Tittarelli, Francesco Pomilio, Luigi Iannetti, Nicola D’Alterio, Mirella Luciani

**Affiliations:** 1Department of Bioscience and Technology for Food, Agriculture and Environment, University of Teramo, Via R. Balzarini 1, 64100 Teramo, Italy; fdonofrio@unite.it (F.D.); apaparella@unite.it (A.P.); 2Istituto Zooprofilattico Sperimentale dell’Abruzzo e del Molise “G. Caporale”, Via Campo Boario, 64100 Teramo, Italy; i.krasteva@izs.it (I.K.); m.tittarelli@izs.it (M.T.); f.pomilio@izs.it (F.P.); l.iannetti@izs.it (L.I.); n.dalterio@izs.it (N.D.); m.luciani@izs.it (M.L.)

**Keywords:** *Listeria monocytogenes*, whole proteome, protein pathogenic, foodborne disease

## Abstract

*Listeria monocytogenes* is a foodborne pathogen that is ubiquitous and largely distributed in food manufacturing environments. It is responsible for listeriosis, a disease that can lead to significant morbidity and fatality in immunocompromised patients, pregnant women, and newborns. Few reports have been published about proteome adaptation when *L*. *monocytogenes* is cultivated in stress conditions. In this study, we applied one-dimensional electrophoresis and 2D-PAGE combined with tandem mass spectrometry to evaluate proteome profiling in the following conditions: mild acid, low temperature, and high NaCl concentration. The total proteome was analyzed, also considering the case of normal growth-supporting conditions. A total of 1,160 proteins were identified and those related to pathogenesis and stress response pathways were analyzed. The proteins involved in the expression of virulent pathways when *L*. *monocytogenes* ST7 strain was grown under different stress conditions were described. Certain proteins, particularly those involved in the pathogenesis pathway, such as Listeriolysin regulatory protein and Internalin A, were only found when the strain was grown under specific stress conditions. Studying how *L. monocytogenes* adapts to stress can help to control its growth in food, reducing the risk for consumers.

## 1. Introduction

*Listeria monocytogenes* is considered as one of the most severe foodborne pathogens, which is responsible for listeriosis, a systemic illness due to the ingestion of contaminated food, such as vegetables, raw meat, milk, and ready-to-eat foods. The clinical manifestations vary from self-limited gastroenteritis with fever, diarrhea, nausea, and vomiting to invasive infections characterized by bacteremia, encephalitis, and fetal loss [1]. Listeriosis has a significant morbidity and a high fatality rate, around 20–30% [2], mainly in immunocompromised individuals, such as cancer patients or those with autoimmune diseases [3], elderly, pregnant women, and newborns [4].

The incidence of listeriosis outbreaks has increased in recent years and several events have been reported in different continents. In the European Union (EU), listeriosis is the fifth most commonly reported zoonosis in humans, with a notification rate of 0.49 cases per 100,000 population in 2021, 14% higher than the rate of 0.43 in 2020. According to the latest available report from EFSA and ECDC [5], a total of 2,183 reported cases of human invasive listeriosis (923 hospitalized and 196 deaths) were observed in the EU, with meat products from bovines or pigs, fruits and vegetables, and sheep’s milk cheeses accounting for the highest values (from 2 to 5%).

Lachmann et al. [6] identified a listeriosis outbreak in health care facilities occurring in Germany between 2014 and 2019, which caused 39 cases and 3 fatal events due to the consumption of meat products. 

Nine hundred cases and 200 deaths were described in South Africa in 2017–2018 and the food vehicle was “polony”, a contaminated processed meat [7].

A total of 253 cases (163 and 90 perinatal and no perinatal patients, respectively) of invasive listeriosis occurred from 2011 to 2016 in 19 provinces of China. Moreover, the authors reported that the highest prevalence (8.9%) was identified for meat/poultry products [8].

McCollum et al. [9] examined 147 outbreak-related cases among 28 states of the United States, mostly caused by Cantaloupe as vehicle source, during 2011. The highest number of patients (127) belonged to people 60 years of age or older (median age of 77 years old), and 33 out of 147 died.

In Taiwan, 338 patients (45.9% over 65 years old) were affected by invasive listeriosis, such as peritonitis, bacteremia, and meningitis, showing an incidence of seasonal variability (more cases in July to September) in the years 2000–2013 [10].

The microorganism is an intracellular Gram-positive bacterium, ubiquitous and largely distributed in nature. *L*. *monocytogenes* can multiply in cold storage, form biofilm by adhering to abiotic surfaces, and resist disinfectants that are normally used in the food industry. Furthermore, it can survive and/or replicate on a wide range of hostile environments, e.g., high salt concentrations, large range of pH and temperatures, and low water activity [11].

The capacity of this pathogen to survive in hostile environments and the virulence potential can vary greatly, depending on the strain [12]. The ability of *L*. *monocytogenes* to colonize food-processing facilities and the formation of persisters of some *L*. *monocytogenes* strains in various niches along the food chain have been described [13,14,15]. Strains that persist for long time in processing environments could provide reservoirs for contamination, ultimately increasing the likelihood of infecting humans. Moreover, interactions between stress response and virulence have been reported in the literature, suggesting that adaptation to adverse environmental conditions (both processing and food environment) could trigger genes involved in virulence [16]. In fact, food-stress-induced adaptive tolerance responses to acid and osmotic stresses can protect the pathogen against similar stresses in the gastrointestinal tract (GIT), thus directly aiding virulence potential [17]. Considering this, it is relevant to understand the mechanisms involved in strain survival and persistence under environmental stresses and in inactivation by the most used disinfectants, and how these could be correlated to virulence in humans.

Whole genome studies of *L*. *monocytogenes* have been used to investigate which pathways are involved in cell adaptation under stress conditions [18]. In general, proteomics encompassed several analytical methods that identify and quantify all proteins in a cell including protein isoforms, post-translational modifications, and interactions among them. This approach provided a post genomic view of a biological system activated as a response to different stimuli [19]. Moreover, proteomics can be used to discover potential biomarkers useful in the understanding of molecular mechanisms and pathways underlying the trait variation in manufactured and stored foodstuffs under different conditions [20]. However, few reports have been published about the expression of pathogenic protein patterns when *L*. *monocytogenes* is exposed to different stress factors [21]. Gel-based proteomic techniques such as one-dimensional electrophoresis and 2D-PAGE combined with mass spectrometry (MS), have been applied to identify which protein patterns were involved in the adaptation of this pathogen to mild acid, alkali stress conditions, as well as high or low temperatures [22]. Tandem mass spectrometry was used to analyze different sub-proteomes of *L*. *monocytogenes*, such as cell wall proteome and secretome. Different protein patterns were found to be expressed, depending on several conditions to which the microorganism was exposed [23,24].

The following study aims to identify which proteins are involved in the expression of virulent pathways when *L*. *monocytogenes* is grown at mild acid, low temperature, and high concentrations of NaCl. All stress conditions were the same as in the previous paper [25], but instead of only immunoproteome, the globally expressed phenotype was analyzed this time. We focused on the different protein patterns expressed during the exponential growth phase and compared them to the optimal growth conditions.

## 2. Materials and Methods

### 2.1. Bacterial Strain and Reference Antisera

*L*. *monocytogenes* ST7 strain (L.M.ST7), as well as positive and negative sera against *L*. *monocytogenes*, were provided by the National Reference Laboratory for *L. monocytogenes* (NRL Lm) of Istituto Zooprofilattico Sperimentale dell’Abruzzo e del Molise “G. Caporale” (Teramo, Italy). This strain was chosen because it was involved in a wide listeriosis outbreak in Italy in 2015–2016 [26].

### 2.2. Listeria monocytogenes’ Growth for Proteomic Analyses

L.M.ST7 (PFGE ApaI 0246 AscI 0356) was grown in brain heart infusion (BHI) broth (Oxoid Thermo Fisher Scientific, Rodano, Italy) and modified BHI broth (pH 5.5, NaCl 7%) at 37 °C and 12 °C according to D’Onofrio et al. [25] in different conditions: C1 (temperature 37 °C, pH 7.0, NaCl 0.5%); C2 (temperature 37 °C, pH 5.5, NaCl 7%); C3 (temperature 12 °C, pH 7.0, NaCl 0.5%); and C4 (temperature 12 °C, pH 5.5, NaCl 7%).

Bacterial cells were collected at the late log phase (optical density at 600 nm). After centrifugation (Eppendorf 5424, Eppendorf, Hamburg, Germany) (5600× *g* at 4 °C for 10 min), the cell pellets were washed three times with 10 mL phosphate-buffered saline solution. All experiments were performed in triplicate.

### 2.3. Protein Extraction and Immunoblotting

Proteins were extracted from the sample using CelLytic B Cell Lysis Reagent and CelLytic IB Inclusion Body Solubilization Reagent (Sigma-Aldrich, Milan, Italy). Subsequently, proteins were precipitated by TCA protocol and solubilized with 0.05% SDS, 1M NaOH. The protein concentration was determined by Pierce^TM^ BCA Protein Assay Kit (Thermo Fisher Scientific, Waltham, MA, USA).

The protein extracts were diluted 1:2 with NuPAGE^TM^ LDS Sample Buffer and NuPAGE^TM^ Reducing Agent (Life Technologies Thermo Fisher Scientific, Monza, Italy) and subjected to agitation at 70 °C for 10 min. The protein band profiles were resolved by loading 12 μg of protein per well using NuPAGE^TM^ 4–12% Bis-Tris pre-cast gels (Life Technologies Thermo Fisher Scientific, Ann Arbor, MI, USA). Immunoblotting was performed using the iBlot™ Gel Transfer Device and iBlot™ Gel Transfer Stacks (Invitrogen Thermo Fisher Scientific, Carlsbad, CA, USA) (1 min at 20 V–1.3 A, 4 min at 23 V–2.1 A, and 2 min at 25 V–1.5 A). Nitrocellulose membrane (Life Technologies Thermo Fisher Scientific) blocking was performed with 10% skim milk (Biolife, Milan, Italy) in PBS 0.05% Tween 20 (PBST). Positive and negative sera against *L. monocytogenes* were diluted 1:5000 in PBST (2.5% skim milk) (PBSTM) and immunocomplexes were identified by anti-sheep IgG (Rabbit) HRP-conjugated antibody (Calbiochem Sigma-Aldrich, Milan, Italy) diluted 1:10,000 in PBSTM. The results were acquired by means of ChemiDoc MP system (Bio-Rad, Hercules, CA, USA).

### 2.4. Mass Spectrometry (MS) and Data Analysis

The proteins were resolved by SDS-PAGE and made visible by Coomassie stain from a 4–12% NuPAGE^®^ precast gel (Thermo Fisher Scientific). Briefly, the gel lanes were cut [27] and reduced/alkylated by 10 mM dithiothreitol (DTT) and 55 mM iodoacetamide (IAA), respectively. The proteins were trypsin digested at 37 °C o/n. Peptides were desalinated and concentrated using StageTip C18, which was applied before peptides nLC-MS/MS analysis [28]. Peptide mixtures were dried under vacuum and solubilized in 5% formic acid. The samples were analyzed by a quadrupole Orbitrap Q-Exactive HF mass spectrometer associated with an UHPLC Easy-nLC 1200 (Thermo Fisher Scientific) combined with a 25 cm—C18 column (75 μm inner diameter) (Dr. Maisch Gmbh, Ammerbuch, Germany). In order to separate peptides, a linear gradient was applied: over 23 min from 95% solvent A consisting of 2% ACN (0.1% formic acid) to 50% solvent B by 80% ACN (0.1% formic acid), and over 2 min from 50% to 100% with a flow rate of 0.25 μL/min with a single run time of 33 min. MS data were obtained in positive mode by a DDA (data-dependent acquisition) top 15 method. Using Orbitrap Q-Exactive HF with 60,000 resolution, automatic gain control (AGC) target 1e6, IT 120 ms, survey full scan MS spectra were acquired. For higher-energy collisional dissociation (HCD) spectra, the resolution was set to 15,000, AGC target 1e5, IT 120 ms, normalized collision energy 28%, isolation width of 3.0 *m*/*z*, and a dynamic exclusion of 5 s.

The nLC-MS/MS method applied and data analysis were carried out according to D’Onofrio et al. [25].

### 2.5. Gene Ontology Analysis

Gene ontology (GO) analysis and classification were performed using ShinyGO v0.76 software [29]. The gene sets identified for each growth condition were used to perform the enrichment analysis. The enriched pathways were identified, and genes were grouped by functional categories. Furthermore, the proteins identified by nLC-ESI-MS/MS analysis were analyzed and summarized by Venn diagram.

### 2.6. STRING Analysis

The open-source STRING software (Search Tool for the Retrieval of Interacting Genes/Proteins) version 11.05 was used to conduct the in-silico analysis. The proteins related to pathogenesis and stress response pathways given by GO analysis were selected in order to identify the protein–protein interactions. A network image was obtained for visualization purposes.

## 3. Results and Discussion

### Proteome Profiling Results

The proteome profiling obtained by nLC-ESI-MS/MS revealed 1,160 proteins, summarized by Venn diagram (Figure 1), with 771, 840, 965, and 650 proteins being identified in C1, C2, C3, and C4 conditions, respectively. Moreover, 64, 64, 144, and 12 unique proteins were identified for the same conditions reported above, respectively. 

In contrast to the optimal condition (C1), the number of genes involved in the pathogenesis pathways was higher compared with the stress response pathways (Table 1). In detail, Table 1 shows which proteins were involved in the pathogenesis and stress response pathways in the different environmental conditions evaluated in this study.

Enolase (Eno) and DNA protection during starvation (Dps) proteins were identified for all experimental conditions. Eno plays a crucial role in the degradation of carbohydrates via glycolysis, and it catalyzes the conversion of 2-phosphoglycerate into phosphoenolpyruvate. Furthermore, Eno binds plasminogen during host infection when it is expressed on the cell wall. This reaction allows *L*. *monocytogenes* to develop surface-associated proteolytic activity, which can contribute to the invasion of tissues and virulence [22]. Eno has also been implicated in the regulation of gene expression of *L*. *monocytogenes*. It seems that Eno binds specific DNA sequences and acts as a transcription factor, regulating the expression of genes involved in a variety of cellular processes, such as general stress response and pathogenesis. In *L*. *monocytogenes*, Eno has been shown to bind to the promoter region of the *hly* gene, which encodes the hemolysin protein. By binding to this region, Eno can modulate the encoding of hly protein [30].

On the other side, Dps protects DNA from oxidative damage [31] and has a role both in resistance to heat and cold shocks and in the virulence pathway, modulating listeriolysin O (LLO) production and its stability. LLO is an essential secreted pore-forming protein that induces vacuole lysis during *L*. *monocytogenes* internalization [32,33]. Dps was involved in other biological processes. For example, it is shown to play a role in regulating iron homeostasis in *L*. *monocytogenes*. Specifically, Dps binds iron ions and regulates their intracellular concentration, including several cellular pathways, such as respiration, metabolism, and stress responses [34]. Moreover, Dps has been implicated in the response of *L. monocytogenes* to various environmental stressors, such as osmotic stress and pH changes. In response to these stressors, Dps could undergo conformational changes that modulate its DNA-binding properties and its ability to protect DNA from oxidative damage [35]. Additionally, it has been suggested to have a role in the formation and maintenance of biofilms [36]. D-alanine-D-alanyl carrier protein ligase (DltA) was identified in all stress conditions; this protein is involved in the lipoteichoic acid (LTA) biosynthesis to preserve cation homeostasis and assimilation, through action on cell wall net charge [37].

One of the important functions of DltA is its role in resistance to cationic antimicrobial peptides (CAMPs). CAMPs are small cationic peptides that are produced by various organisms as part of their innate immune system. They can disrupt bacterial cell membranes and are an important defense mechanism against bacterial infections. DltA plays a crucial role in *L*. *monocytogenes* resistance to CAMPs by modifying the net charge of the cell wall. In particular, DltA catalyzes the transfer of D-alanine-D-alanine dipeptides to the carrier protein DltC, which then transfers the D-alanine-D-alanine to LTA. This modification reduces the net negative charge of the cell wall and makes it less susceptible to CAMPs. Moreover, DltA has been implicated in the regulation of virulence in *L*. *monocytogenes*. Specifically, DltA has been also implicated in regulating the expression of virulence genes by modulating the activity of the transcriptional regulator “Listeriolysin positive regulatory protein” (PrfA) [38].

PrfA, encoded by *prfA* gene, was found only in acidic and osmotic stress conditions; this is a very interesting finding, as this protein has been pointed out in the literature as a “master regulator of virulence” [16]. The lack of PrfA greatly attenuates the virulence of *L*. *monocytogenes*, in both vertebrates and invertebrates, as well as in cell culture infection models [39,40,41]. This protein is also known as a positive regulator of listeriolysin, 1-phosphatidylinositol phosphodiesterase (PI-PLC), and several pathogenic factors [42]. Moreover, *prfA* gene seems to be related to the adaptative response to acid stress of *L*. *monocytogenes*. The production of this protein only in the presence of both osmotic and acidic stress conditions confirms the link between this protein and stress-resistant cells, similarly to what was recently reported for oxidative stress resistance [43]. On this basis, *L*. *monocytogenes* strains able to survive/grow in foods with low pH and high salt concentration could be potentially considered as more virulent. Therefore, more attention should be paid to strains isolated in these types of food (e.g., dried meat products or long-aged cheeses, as well as foods containing acidic preservatives), in which the pathogen is often exposed to severe osmotic and acid stresses, even at the same time. The presence of food preservatives in the gastrointestinal tract, particularly organic and inorganic acids, is an important hurdle for *L*. *monocytogenes*. In fact, organic acid dissociation in the cell cytoplasm implies acidification and proton/anion influx, which inhibits both transport of catabolites and ATP synthesis, leading to cell membrane, nucleic acid, and protein damages. To face acidic stress, *L*. *monocytogenes* genome acts by a wide spectra of gene expression changes, such as *rB*, *CtsR*, and *PrfA* involved in DNA repair, fatty acid biosynthesis related to cell wall modification, and oxidative stress defense [44].

Motility gene repressor (MogR) and Internalin A (InlA) proteins were identified in low temperature conditions, C3 and C4, respectively. The former induces transcriptional repression of flagellar motility genes at 37 °C and during infection [45], whereas the latter acts as mediator of *L*. *monocytogenes* entrance into host intestinal epithelial cells, binding the cadherin-1 receptor (E-cadherin, CDH1) in human subjects. This could explain the potential role of storage of foods at a low temperature in favoring the production of these proteins highly involved in the pathogenesis of listeriosis. Recently, the joint effects of low temperature and osmotic stress were evaluated in relation to the capacity of *L*. *monocytogenes* to invade human intestinal CACO-2 cells, and the results were strain-dependent [46]. In our study, these proteins associated with cellular invasion during infection were identified only under stressful conditions, namely at a low temperature, in both the presence (C4) and absence (C3) of osmotic stress, and thus shown to be mostly related to cold stress rather than to high salt concentration.

The list of all proteins identified for each condition and GO enrichment analysis results are available as Appendix A. As for the genes grouped under the “stress response” category, listed here below, damage control and DNA-repair-associated genes were expressed for all conditions, shown to be not strictly related to the presence of stressful environmental conditions.

Single-stranded DNA-binding (Ssb1), DNA repair (RadA), and RecA proteins have a key role in DNA replication, recombination, and repair [47].

In addition, Ssb1, RadA, and RecA proteins have been found to be involved in various other biological processes in *L*. *monocytogenes*. Ssb1, for example, influenced stress response and virulence; it was upregulated in response to heat shock and oxidative stress, and its expression was required for the full virulence of *L*. *monocytogenes* in a mouse model of infection. Additionally, Ssb1 has been implicated in cell wall biosynthesis and antibiotic resistance. On the other hand, RadA has been found to be important for the survival of such bacteria under DNA-damaging conditions [48]. Specifically, RadA was required for the repair of DNA double-stranded breaks, which can be caused by exposure to ionizing radiation or other DNA-damaging agents. Moreover, RadA was shown to be important for the formation and maintenance of *L*. *monocytogenes* biofilms. Finally, RecA protein has been implicated in the regulation of bacterial virulence. Specifically, RecA regulated the expression of virulence genes by modulating transcriptional regulator *PrfA* activity. Additionally, RecA has been implicated in the regulation of stress, including responses to heat shock and DNA damage [49].

Endonuclease MutS2, DNA mismatch repair MutS, and DNA mismatch repair MutL proteins repair mismatches in DNA. Furthermore, MutS2 seems to have a role in bacterial genetic diversity control, acting as a small molecule with sensor activity [50]. In addition to their roles in DNA repair and maintenance of genetic stability, the endonuclease MutS2, MutS, and MutL proteins have been found to be involved in various other biological processes in *L*. *monocytogenes*. MutS2 has been shown to play a role in the regulation of bacterial genetic diversity in such bacteria; specifically, MutS2 had ribonuclease activity and cleaved small non-coding RNAs, leading to changes in gene expression. MutS2 had an important significance for the adaptation of such a pathogen to different environments and for the regulation of its virulence gene expression. Moreover, it seemed to modulate *L*. *monocytogenes* full virulence in a mouse model of infection, and its expression was upregulated in response to various stresses, including low pH, salt stress, and exposure to bile salts. MutL has been implicated in the regulation of biofilm formation of the pathogen. Specifically, it was required for robust biofilms formation, and its expression was upregulated in response to nutrient limitation [51].

DEAD-box ATP-dependent RNA helicase (CshB) is involved in heat and cold tolerance, oxidative and alkali stress, and motility [52,53].

RNA helicases are characterized by the specific amino acid sequence D-E-A-D (Asp-Glu-Ala-Asp) in the conserved helicase motif II. They are fundamental for RNA secondary structure resolution under cold stress conditions, allowing transcription and translation events [54]. In addition to its roles in stress tolerance and motility, CshB was involved in the regulation of gene expression. It seemed to interact with a number of different RNA molecules, including small regulatory RNAs and mRNAs encoding virulence factors. By unwinding RNA secondary structures and promoting RNA degradation or stabilizing RNA structures, CshB could influence the levels of these RNAs and thereby modulate gene expression. Furthermore, CshB was required for *L*. *monocytogenes* full virulence and its expression was upregulated in response to various stresses, including low pH and oxidative stress [53].

Chaperone DnaJ and DnaK proteins are produced to respond to hyperosmotic and heat shock. They prevent the formation of stress-denatured proteins by means of ATP-dependent interactions, leading to efficient folding [55]. Chaperone protein works with DnaJ and DnaK. It is involved in cell recovery pathways induced by heat and osmotic damage [56,57]. Moreover, ClpB contributes to *L*. *monocytogenes* virulence, acting as a chaperone [58]. The *clpB* gene is related to defective cellular growth when such a microorganism is exposed to NaCl osmotic stress [59,60]. For this reason, chaperone proteins are considered to be of crucial importance to restore protein structures and, consequently, their functions are vitiated by osmotic stress [61]. In addition to their roles in responding to heat and osmotic stress, chaperones DnaJ, DnaK, and ClpB orchestrated *L*. *monocytogenes* virulence: DnaJ and DnaK were fundamental for the proper folding and function of the bacterial surface protein InlA [60], while ClpB was fundamental for Actin A (ActA), which is a factor involved in bacterial movement and spreading within host tissues [62].

ATP-dependent helicase/nuclease subunit A and B (ADDA and ADDB) contribute to DNA repair and recombination pathways as UvrABC system proteins (UvrA, UvrB, and UvrC) and RuvA–RuvB complex [50]. In addition, ADDA and ADDB were implicated in the regulation of *L*. *monocytogenes* virulence; specifically, their expression was significantly upregulated in *L*. *monocytogenes* strains that were more virulent than others.

Moreover, it seemed that the deletion of the genes encoding ADDA and ADDB resulted in decreased virulence in a mouse model of *L*. *monocytogenes* infection [62].

The RecF and lexA proteins act in tandem and are involved in the replication of DNA and basal SOS response, both with RecA. Pyrophosphatase (Ppax) plays a role in DNA repair, modulating the intracellular pyrophosphate pool [49]. The stress response category also includes genes involved in oxidoreductase activity, such as peptide methionine sulfoxide reductase (msrB) and thioredoxin reductase (trxB), which contribute to cell redox homeostasis, removing superoxide radicals [63]. Among all the “stress response pathway” proteins, only MsrB and PpaX were specific to certain stress conditions. Specifically, MsrB was expressed only in the presence of both osmotic and acidic stress, while PpaX was found only in response to cold stress.

STRING analysis highlighted a network characterized by 28 nodes and 116 edges (Figure 2). The cluster indicates that the proteins are biologically connected and involved in stress response regulation and pathogenesis [64].

The production of certain proteins, particularly those involved in pathogenesis, such as listeriolysin regulatory protein and InlA, can be induced by stress conditions as a mechanism of the adaptation and survival of the bacteria. These proteins may help *L*. *monocytogenes* to overcome stressors in the environment and continue to grow and cause infection. Furthermore, Manso et al. [65] found that many stress response genes in *L*. *monocytogenes* are regulated by SigB, which allows the pathogen to respond to stress by activating the expression of genes involved in stress tolerance and adaptation. Moreover, Sibanda et al. [17] showed that *L*. *monocytogenes* can adjust its gene expression to adapt to various stress conditions encountered during foodborne transmission. This mechanism is regulated by the crosstalk between SigB and PrfA. SigB plays a crucial role in the stress response by controlling gene expression that aids the survival of the bacteria in difficult environmental conditions, i.e., shifts in osmotic pressure, temperature, pH, redox potential, and nutrient availability [66].

## 4. Conclusions

Elucidating the mechanism of *L*. *monocytogenes* stress response implies knowledge about the proteins involved in such pathways and how their expression is regulated. In this respect, the results of this study may be utilized to understand *L*. *monocytogenes* metabolism when it is exposed to different stress factors and how these conditions can induce the production of specific proteins involved in the pathogenic pathways. Overall, stress response entailed different pathways, but some proteins seemed to be associated with stress conditions, namely osmotic, acidic, and cold stress. These findings indicate the usefulness of a deep knowledge of the proteomic characteristics of different *L*. *monocytogenes* strains, as their response could vary and be linked to the peculiar environmental conditions encountered during food production and storage.

The presence of certain proteins in the proteome, particularly those included in the “pathogenic pathway”, is the final proof of the pathogenicity of certain isolates, which cannot be based only on the presence of the codifying gene. The characterization of the other proteins detected in this study by means of bioinformatic tools (i.e., VirulentPred and Vaxijen) is already in progress, with the aim of identifying the potential immunogenic proteins involved in the virulence pathways. Future investigations might involve the development and optimization of a CRISPR-Cas-assisted recombineering system to facilitate bacterial genetic manipulation [67]. Using this system, point mutations, deletions, insertions, and gene replacements can be easily generated on the chromosome or native plasmids in *L*. *monocytogenes*. In this way, the pathogenic role of these proteins could be confirmed.

## Figures and Tables

**Figure 1 foods-12-02166-f001:**
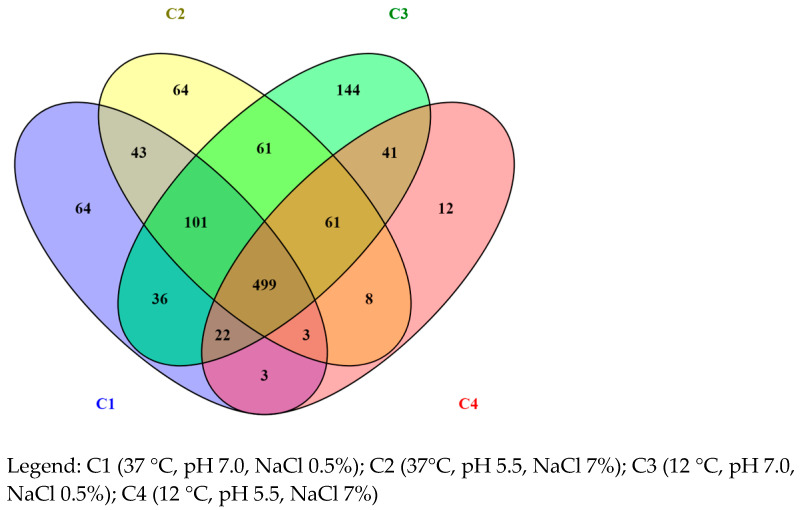
List of proteins summarized by Venn diagram.

**Figure 2 foods-12-02166-f002:**
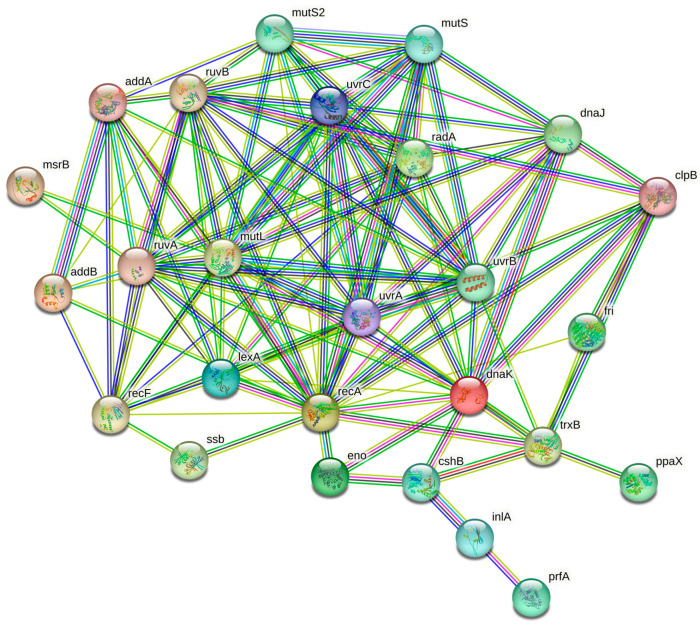
A network diagram created using STRING v11.05 to show the protein–protein interactions of upregulated proteins in *L. monocytogenes* after exposure to C2, C3, and C4 environmental stress conditions. Nodes represent proteins, while edges indicate their interactions. The edges are color-coded to represent different types of interactions, including known (pink and light blue), predicted (green, red, and blue), and other (yellow, black, and grey) interactions.

**Table 1 foods-12-02166-t001:** List of proteins involved in the “pathogenesis” and “response to stress” pathways.

Gene	Entry	Protein Name	Length (AA)	Condition
**Pathogenesis**				
*eno*	P64074	Enolase	430	C1, C2, C3, C4
*dps*	Q8Y8G1	DNA protection during starvation protein	156	C1, C2, C3, C4
*dltA*	Q8Y8D4	D-alanine--D-alanyl carrier protein ligase	510	C2, C3, C4
*prfA*	P22262	Listeriolysin regulatory protein	237	C2
*mogR*	P0DJO8	Motility gene repressor MogR	306	C3
*inlA*	P0DJM0	Internalin A	800	C4
**Response to stress**				
*ssb1*	Q8YAR8	Single-stranded DNA-binding protein	178	C1, C2, C3, C4
*radA*	Q48761	DNA repair protein	457	C1, C2, C3, C4
*mutS2*	Q8Y7P1	Endonuclease MutS2	785	C1, C2, C3, C4
*recA*	P0DJP0	Protein RecA	348	C1, C2, C3, C4
*mutS*	Q8Y789	DNA mismatch repair protein MutS	860	C1, C2, C3, C4
*mutL*	Q8Y788	DNA mismatch repair protein MutL	601	C1, C2, C3, C4
*cshB*	Q8Y755	DEAD-box ATP-dependent RNA helicase CshB	435	C1, C2, C3, C4
*dnaJ*	P0DJM1	Chaperone protein DnaJ	377	C1, C2, C3, C4
*dnaK*	P0DJM2	Chaperone protein DnaK	613	C1, C2, C3, C4
*ruvB*	Q8Y6Z8	Holliday junction ATP-dependent DNA helicase RuvB	335	C1, C2, C3, C4
*clpB*	Q8Y570	Chaperone protein ClpB	866	C1, C2, C3, C4
*addA*	Q8Y511	ATP-dependent helicase/nuclease subunit A	1235	C1, C2, C3, C4
*addB*	Q8Y510	ATP-dependent helicase/deoxyribonuclease subunit B	1157	C1, C2, C3, C4
*trxB*	O32823	Thioredoxin reductase	319	C1, C2, C3, C4
*uvrA*	Q8Y4F6	UvrABC system protein A	956	C1, C2, C3, C4
*uvrB*	Q8Y4F5	UvrABC system protein A	658	C1, C2, C3, C4
*ruvA*	Q8Y6Z7	Holliday junction ATP-dependent DNA helicase RuvA	201	C1, C2, C3
*recF*	Q8YAV8	DNA replication and repair protein RecF	370	C2, C3, C4
*uvrC*	Q8Y7P0	UvrABC system protein C	603	C2, C3, C4
*lexA*	Q8Y7H7	LexA repressor	204	C1, C3, C4
*msrB*	Q8Y641	Peptide methionine sulfoxide reductase MsrB	145	C2
*ppaX*	Q8Y4G3	Pyrophosphatase PpaX	217	C3

Legend: AA = amino acid; C1 (37 °C, pH 7.0, NaCl 0.5%); C2 (37°C, pH 5.5, NaCl 7%); C3 (12 °C, pH 7.0, NaCl 0.5%); C4 (12 °C, pH 5.5, NaCl 7%).

## Data Availability

The data presented in this study are available on request from the corresponding author.

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
