# Peer review of "Stress Adaptation Responses of a *Listeria monocytogenes* 1/2a Strain via Proteome Profiling"

_foods, 2023, doi:10.3390/foods12112166_

Round 1

Author Response

The authors thank the reviewer for its positive and thorough reviews. In the revised text all changes are highlighted in yellow.

General comments:

Authors assessed which proteins were involved in the expression of virulent pathways in a Listeria monocytogenes strain under the stresses including low acid, or low temperature, or high salt environments, compared with normal growing conditions, using 1D electrophoresis, and 2D PAGE with MSn. 1,160 proteins reported to be associated with pathogenesis and stress response. Under certain stress, proteins associated with Listeriolysin and Internalin A were exclusively found.

The introduction went into great length justifying of why L. mono is bad and important, but the background on proteomics, currently available tools and methods, current stage of pathogen’s proteomics studies, seemed short and out of proportion in comparison. The discussion is mostly comprised of descriptions and explanations on select proteins and pathways.

Thanks for your constructive comment. The authors improved the significance and the presentation of proteomic accordingly.

Other than some minor English usage/ styling, and spots where it seemed citations were missing, the reviewer does not have much to comment on the authors’ discussions.

Specific comments:

Check missing verbs and articles: i.e. “…only in presence…” “…only in the presence…”; “…similarly to what recently…”  “…similar to what was recently…”

We modified as suggested.

17-19              The description of the control can be more concise.

We improved accordingly.

19                    “1 160” styling. Suggest “1160” or “1,160”.

You are right. We modified it.

39                    (14% higher… 2020) can be place at the end of the sentence without parenthesis.

We changed as suggested.

Section 2.1      Suggest placing origins of the chemical reagents in parenthesis right after the first mention of the reagent.

We added chemical reagents’ information and so we deleted the first sentence regarding the origin of all used reagents.

180                  Suggest “is shown” instead of “showed”

We modified as suggested

191-198          Possible missing citations.

We added the missing citation. 194-205 have same citation with 35 as the number. 

Reviewer 2 Report

Manuscript title:

Stress adaptation responses of a Listeria monocytogenes 1/2a 2 strain via proteome profiling

Review Comments

The manuscript titled “Stress adaptation responses of a Listeria monocytogenes 1/2a strain via proteome profiling” is interesting and has scope and applications. However, the present form of the manuscript is not satisfactory in terms of the content and overall presentation. There re substantial concerns evident in the present manuscript. The authors are interested to revise the manuscript, the following comments may helpful.

There are no technical details for the methodology "Protein extraction and immunoblotting" Stress adaptation

The authors mentioned that "Certain proteins, particularly those involved in the pathogenesis pathway such as Listeriolysin regulatory protein and Internalin A, were only found when the strain was grown under specific stress conditions". However, the authors have not clearly discussed the relationship between the production of these proteins and specific stress conditions. Why the proteins are producing under these stress conditions? Did you try the conditions already described in the literatures?  How the stress adaptation happens and their impact in the virulence.

Further, the authors stated that "A total of 1160 proteins were identified, which were associated with the pathogenesis and stress response pathways" Then why the authors selected only limited number of proteins for stress adaptation studies as mentioned earlier?

The novelty and uniqueness of the study is not clear and how the present study mitigated the gaps existed in the literature and what new knowledge is contributed to the readers?

It is suggested that the authors should perform some more studies to vigorously analyse the proteome after MS analysis using some relevant computational biology/ Bioinformatics/ Mathematical modelling to provide some fascinating data about the stress adaptation responses of a Listeria monocytogenes by proteome profiling approaches

There are no interesting figures that provide the outcome of the study. There should be more figures on the potential outcome/data of the present study and subsequent suggested computational biology analysis.

Reviewer 3 Report

Dear authors, 

In my opinion, the recommendation is that minor revisions are necessary. The subject addressed in this article is relevant and worthy of investigation. It is well written and based on the data presented.

Topic 2.2 - line 114 - rewrite the sentence " After centrifugation (5600xg at 4°C for 10 min), the cell pellets were washed three times with 10 ml of phosphate-buffered saline solution."

Include in topic 2.4. the model and manufacturer of the equipment used in the analysis.

For this study you chose a specific sample of ST and determined pulse type (L.M.ST7 (PFGE ApaI 0246 AscI 0356). It would be important to explain why this strain was chosen and not compared with a reference strain.

Author Response

The authors thank the reviewer for its positive and thorough reviews. In the revised text all changes are highlighted in yellow.

Dear authors, 

In my opinion, the recommendation is that minor revisions are necessary. The subject addressed in this article is relevant and worthy of investigation. It is well written and based on the data presented.

Topic 2.2 - line 114 - rewrite the sentence " After centrifugation (5600xg at 4°C for 10 min), the cell pellets were washed three times with 10 ml of phosphate-buffered saline solution."

Thanks for your suggestion. We modified accordingly.

Include in topic 2.4. the model and manufacturer of the equipment used in the analysis.

We agree with this comment and add information as requested.

For this study you chose a specific sample of ST and determined pulse type (L.M.ST7 (PFGE ApaI 0246 AscI 0356). It would be important to explain why this strain was chosen and not compared with a reference strain.

The authors specified why such strain was studied. In particular, the strain caused a listeriosis outbreak in Italy in the years 2015-2016.
